# Performance Analysis of Moving Target Shadow Detection in Video SAR Systems

**Boxu Wei, Anxi Yu \*, Wenhao Tong and Zhihua He**

College of Electronic Science and Engineering, National University of Defense Technology (NUDT), Changsha 410073, China; wbx@nudt.edu.cn (B.W.); twh10355@nudt.edu.cn (W.T.); zhihuahe@nudt.edu.cn (Z.H.)
\* Correspondence: yu_anxi@nudt.edu.cn; Tel.: +86-137-8731-7450

**Abstract:** The video synthetic aperture radar (ViSAR) system can utilize high-frame-rate scene motion target shadow information to achieve real-time monitoring of ground mobile targets. Modeling the characteristics of moving target shadows and analyzing shadow detection performance are of great theoretical and practical value for the optimization design and performance evaluation of ViSAR systems. Firstly, based on the formation mechanism and characteristics of video SAR moving target shadows, two types of shadow models based on critical size and shadow clutter ratio models are established. Secondly, for the analysis of moving target shadow detection performance in ViSAR systems, parameters such as the maximum detectable speed of moving targets, the minimum clutter backscatter coefficient, and the number of effective shadow pixels of moving targets are derived. Furthermore, the shadow characteristics of five typical airborne/spaceborne ViSAR systems are analyzed and compared. Finally, a set of simulation experiments on moving target shadow detection for the Hamas rocket launcher validates the correctness and effectiveness of the proposed models and methods.

**Keywords:** video synthetic aperture radar; moving target; shadow detection; performance analysis

## 1. Introduction

Video synthetic aperture radar (ViSAR) technology integrates the advantages of high-resolution imaging and video imaging while enabling all-weather, all-day observation of dynamic targets [1–3]. Compared to traditional SAR methods, ViSAR is more advantageous for interpreting images over time. When the radar energy is obstructed by targets of a certain height, a shadow area, known as a shadow, is formed. In SAR images, the size and intensity of target shadows can be used for target detection, positioning, tracking, and identification. Utilizing the shadows of dynamic targets in ViSAR temporal images can provide high-precision information on target positions, velocities, and other motion states, making it an important radar observation parameter. This information can be calculated based on the physical size and mobility of the target, platform and radar imaging parameters, and scene clutter intensity.

Detecting targets based on dynamic target shadows is a cutting-edge research topic with significant research value [4–7]. There are two advantages to detecting targets based on dynamic target shadows: firstly, when a moving target has a velocity component in the range direction, there will be a Doppler frequency shift, causing the energy of the moving target to shift from its actual position. Secondly, as the radar wave loses illumination, the actual moving target will exhibit distinct shadow characteristics [8,9]. Compared to traditional SAR-GMTI methods, shadow-based detection methods offer higher positioning accuracy, better performance, and a lower minimum detectable velocity [10].

However, this work also faces some unresolved technical challenges. Firstly, in ViSAR images, shadows are often difficult to distinguish from low-scattering areas such as roads. Methods using shadow intensity for detection may lead to a large number of false alarms.

Secondly, due to the unstable backscatter of video SAR image sequences, false alarms are often generated in complex backgrounds. To address these issues, ref. [11] derived theoretical formulas for the size and intensity of the shadows of arbitrary moving targets in SAR images. They combined median filtering algorithms to process SAR images to obtain higher shadow clutter ratios, classified the detection probability and false alarm probability of the filtered images, and ref. [12] analyzed the relationship between shadow area detection performance and parameters such as road electromagnetic scattering characteristics, clutter environment, and noise amplitude. They combined shadow detection with road auxiliary information to detect and estimate the parameters of moving targets. Furthermore, to improve the distinction between foreground and background, ref. [13] proposed a complex background suppression method based on background, target, and noise decomposition. Ref. [14] realized ViSAR image coherence change detection based on superimposed average coherence. Ref. [15] initially applied deep learning to the field of image denoising, reducing the impact of speckle noise in video SAR signals through the RED20 deep learning network model. This model effectively controls the influence of noise on target detection. Yang et al. [10] studied a method for focusing on ground-moving targets in ViSAR to obtain clearer dynamic target shadows and improve the detection capabilities of the ViSAR system. Ding et al. [16] combined deep learning with sliding window density clustering, bi-long short-term memory networks, and fast region convolutional neural networks to achieve shadow tracking. In the previous literature, traditional methods such as [12–14] utilized parameters like information entropy and shadow contrast to describe the ViSAR image processed by the algorithm. They also employed metrics such as correct detection count, false alarm count, and missed detection count, combined with ROC curves and PR curves, to assess the algorithm's shadow detection capability. Meanwhile, methods like [5,6,10,15] supplemented these by introducing additional metrics like F1 score and map, enabling a comprehensive evaluation of detection algorithm performance. These evaluation methods primarily focus on the statistical analysis of the processed sample data. Furthermore, refs. [11,16] took a step further by integrating theoretical analysis with probability of false alarm and probability of detection to describe shadow detection capability, elucidating the relationship between detection performance and motion speed. However, the aforementioned literature lacks quantitative estimation models for the shadow detection capability of a given ViSAR system, starting from parameters describing shadow characteristics.

To better utilize the shadow features of ViSAR moving targets in algorithms, we modeled the shadow characteristics of moving targets to obtain formulas for shadow size and intensity. By analyzing the impact of target parameters, background parameters, and system parameters on detection performance in ViSAR moving targets, we estimated the speed of the target under a given system and validated the results through simulation experiments. The conclusions of this paper can support the design of ViSAR radar systems and shadow-based moving target detection tasks.

## 2. Mechanisms of ViSAR Moving Target Shadow Formation and Shadow Models

### 2.1. Formation of Two Types of Moving Target Shadows

In ViSAR sequences, ground vehicles and other moving targets exhibit distinct shadow characteristics. Figure 1 shows a single-frame video image from a ViSAR video released by Sandia National Laboratories (SNL), demonstrating the shadow characteristics of ground-moving vehicles in a real ViSAR image. When the vehicle stops, the target energy is concentrated. When the target moves, defocusing occurs, causing the target energy to deviate from its actual position. Additionally, due to the electromagnetic wave obstruction of moving vehicles on the ground, a distinct shadow feature is formed at the actual position of the target, indicating the actual position of the vehicle. The shadow trajectory of the moving vehicle along the road in Figure 1 is the actual running trajectory of the vehicle target, which can be used for the detection and tracking of ground-moving targets. The shadow characteristics of moving targets are closely related to target size, speed, ground clutter scattering characteristics, and SAR system parameters.

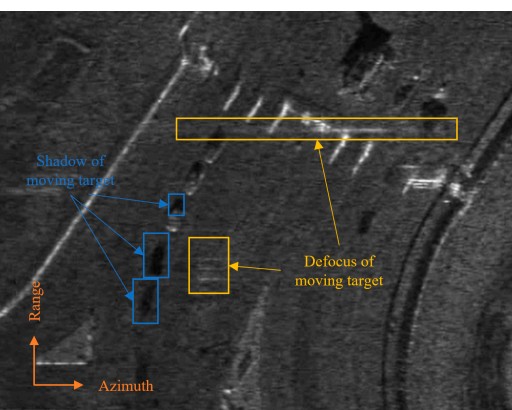

**Figure 1.** Shadow characteristics of ground-moving vehicles in real ViSAR images.

Referring to reference [5], this paper illustrates the mechanism of shadow formation and shadow geometry of ViSAR targets in Figure 2. Considering the radar operating in the geometry of normal side view, let the viewing angle be $\alpha$, the synthetic aperture time be $T_a$, and there are three vehicle targets in the scene, all assumed to be rectangular structures, as shown in green in Figure 2a, with the width and height of the targets being $W_T$ and $H_T$, and lengths of $L_{T1}$, $L_{T2}$, and $L_{T3}$ respectively. Target 1 is a stationary target, while targets 2 and 3 are moving at a speed $V_T$ on a uniform clutter background with a motion direction angle of $\theta$ with respect to the range direction, moving from the green part to the position within the dashed line box. The moving targets move the same distance within the synthetic aperture time, which is $V_T T_a$ ($L_{T3} > V_T T_a > L_{T2}$).

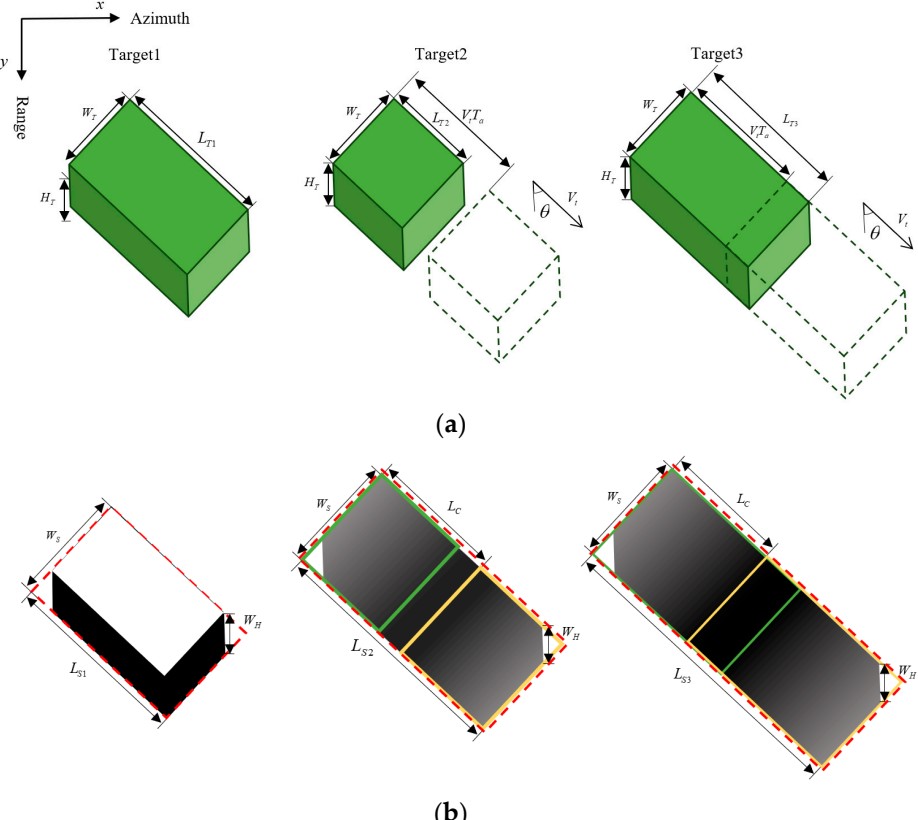

**Figure 2.** Mechanism of shadow formation and shadow geometry of ViSAR targets. (**a**) The geometric models of the three targets. (**b**) The corresponding shadows of the targets in (**a**).

Figure 2b corresponds to the shadows generated by the three targets shown in Figure 2a. The stationary target 1 and its shadow in the SAR image are depicted as a small, pure shadow area at the far end of the range direction, caused by the elevation obstruction of the target. When the target is in motion, a mismatch of the matched filter during the imaging process can result in defocusing, while a Doppler frequency difference can cause an energy shift in moving targets. This results in the target shadow not being obscured. Additionally, as the duration of shadowing varies at different positions within the shadow zone and the coherent accumulation time varies, the energy within the shadow area changes with location. Comparing targets 2 and 3, due to their different sizes, the larger target 3's central region is completely obscured, forming a pure shadow area in the middle. The shadow region can be approximated using the red-dash rectangle in the image. In Figure 2b, the green and yellow boxes represent the positions of the target before and after movement during the synthetic aperture time. We controlled the speeds of targets 2 and 3 to be the same while in motion, resulting in the same displacement $L_C$ during the synthetic aperture time. The shadow region of a moving target is composed of two parts: the area where the target itself is located and the area where it is occluded by elevation. Compared to the shadow of a stationary target, the shadow size of a moving target significantly increases. The shadow size along the range direction $W_H$, caused by elevation obstruction of the target can be calculated from Equation (1), and further calculations can determine the width and length of the shadow region of a moving target, denoted as $W_S$ and $L_S$ respectively in Equation (2).

$$W_H = H_T \times \tan \alpha \tag{1}$$

$$\begin{cases} W_S = W_T + W_H \times \sin \theta = W_T + H_T \times \tan \alpha \times \sin\theta \\ L_S = L_C + L_T + W_H \times \cos \theta = L_C + L_T + H_T \times \tan \alpha \times \cos \theta \end{cases} \tag{2}$$

In Equation (2), $L_C$ represents the distance moved by the target during the synthetic aperture time, and $L_T$ represents the length of any target along the direction of movement. Additionally, the shadow area can be calculated as $S_S$ in Equation (3).

$$S_S = W_S \times L_S \tag{3}$$

Next, let us analyze the intensity characteristics of the shadow region of moving targets. From Figure 2, it can be observed that the target shadow exhibits a gradual intensity change trend of "light gray–dark gray–light gray" along the motion direction, with a gradual transition shadow region at each end and a stable intensity shadow region in the middle. Additionally, it can be observed that under the same motion speed, the intensity characteristics of the shadows of targets 2 and 3 in the ViSAR image differ due to the difference in target scales. The stable shadow region of target 2 is a non-pure shadow region, appearing as dark gray, while the stable shadow region of target 3 is a pure shadow region, appearing as black.

The shadow formed by moving target 2 is referred to as a Type I shadow, and the shadow formed by target 3 is referred to as a Type II shadow. Figure 3 illustrates the formation of the two types of shadows, with the horizontal axis representing the position of scattering points along the motion direction of the target shadow and the orange segment corresponding to the time interval when the point is occluded, not exceeding the synthetic aperture time. The lower images in Figure 3 show the formation effects of the two types of shadows by moving targets on a uniformly cluttered background. The longer the yellow segment, indicating a longer occlusion time within the synthetic aperture time, the lower the corresponding SAR image grayscale value.

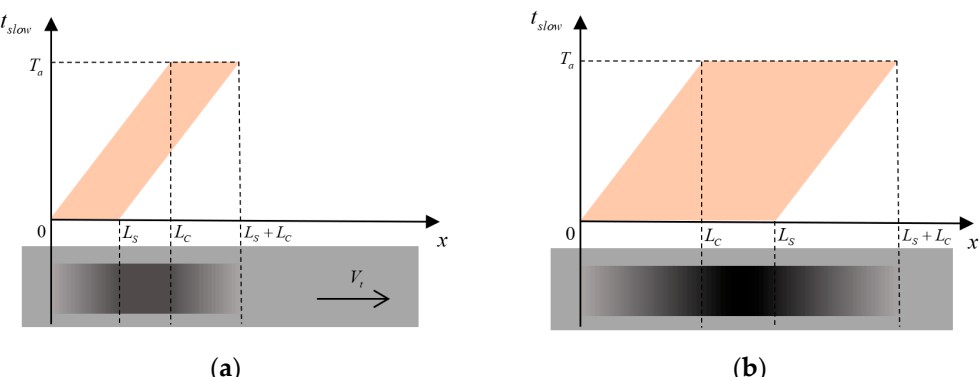

**Figure 3.** Schematic diagram of ViSAR dynamic target shadow formation. (**a**) Type I shadow, $L_T \leq L_C$. (**b**) Type II shadow, $L_T > L_C$.

Let $\lambda$ be the radar signal wavelength, $R_0$ be the slant range, $\rho$ be the azimuth resolution, and $V$ be the platform velocity, then $T_a$ can be calculated using Equation (4).

$$T_a = \frac{\lambda R_0}{2\rho V} \tag{4}$$

The critical size $L_C$ can be expressed as the product of the target velocity and the synthetic aperture time, as shown in Equation (5).

$$L_C = V_T T_a = \frac{V_T \lambda R_0}{2\rho V} \tag{5}$$

Comparing the two models in Figure 3, it is found that Type I shadow and Type II shadow can be distinguished based on the relationship between $L_C$ and $L_T$ and the value of $L_C$ remaining constant under the same target motion speed.

(1)    Type I shadow, $L_T \leq L_C$

At this point, the target length is less than the critical size, which is smaller than the displacement of the target within the synthetic aperture time. The stable shadow area in the middle of the target shadow is occluded for a time shorter than the synthetic aperture time, showing a segment of dark gray non-pure shadow area with a length of $L_C - L_T$, as shown in Figure 3a.

(2)    Type II shadow, $L_T > L_C$

At this point, the target length is greater than the critical size, and the stable shadow area in the middle of the target shadow is always occluded and not illuminated by radar waves, showing a segment of black pure shadow area with a length of $L_T - L_C$, as shown in Figure 3b.

Analysis shows that the smaller $L_C$ is relative to the target length, the easier it is to form a Type II shadow. In ViSAR moving target detection, due to the larger intensity difference of the Type II shadow compared to the surrounding clutter area, it is usually easier to detect. By analyzing the various influencing factors of the critical size in Equation (5), at higher frequency bands, lower resolution ViSAR systems, or when the target motion speed is relatively small compared to the platform speed, or when the detection distance is closer, the critical size of the target will be smaller, making it easier to form a Type II shadow accordingly.

On the other hand, the larger the area of the moving target shadow region, the more pixels in the shadow region, which is beneficial for the overall improvement of the detection performance of the entire moving target. According to Equation (2), the length of the shadow area is approximately the sum of the target length and the critical size. The larger the target size and critical size, the more pixels in the shadow area. Therefore, considering the formation of Type II shadows and the contribution of the number of shadow area pixels

to the detection of moving target shadows, when the target has a moderately sized critical size, better detection performance is expected.

### 2.2. Signal-to-Clutter Ratio of Moving Target Shadows

SAR receives thermal noise, and its coherent imaging mechanism inevitably introduces additive noise and multiplicative noise into SAR images [9,17]. The total noise in SAR images is the square root of the sum of additive noise and multiplicative noise. Considering that the moving target falls in a uniformly cluttered background, we define the total noise of the radar system as $\sigma_N$, represented by Equation (6).

$$\sigma_N = \sigma_n + MNR\overline{\sigma_b} \tag{6}$$

where $\sigma_n$ represents the additive noise equivalent backscatter, and the multiplicative noise is proportional to the average scene intensity of the clutter scene. It is represented by the product of the scene average backscatter $\overline{\sigma_b}$ and multiplicative noise ratio ($MNR$). Based on this, for a coherent speckle background, the scattering intensity of the clutter scene and the shadow region are defined as $\sigma_C$ and $\sigma_S$, respectively, as shown in Equation (7), where the intensity is the power of the pixel value or the square of the magnitude of the image.

$$\sigma_C = \sigma_N + \sigma_b = \sigma_n + MNR\overline{\sigma_b} + \sigma_b \tag{7}$$

In the equation, $\sigma_b$ represents the scene backscatter coefficient, and the selection of $\sigma_b$ is related to the radar's echo power. The reflectivity coefficient is linearly related to power [9]. In this paper, due to the selection of a uniform clutter scene, $\sigma_b$ is approximately equal to $\overline{\sigma_b}$.

During the imaging process of video SAR, the coherent accumulation loss of ground scatter points within the synthetic aperture time caused by the occlusion of moving targets results in shadows with weaker image intensity compared to the surrounding areas. Due to target occlusion, the scattering intensity of the moving target shadow region $\sigma_S$ can be defined as follows:

$$\sigma_S = \sigma_N + \sigma_s = \sigma_n + MNR\overline{\sigma_b} + (1 - \beta)\sigma_b \tag{8}$$

In the equation, $\beta$ represents the shadow coefficient, indicating the proportion of time that the ground is occluded by moving targets within the synthetic aperture time, $0 \leq \beta \leq 1$. The larger $\beta$, the weaker the scattering in that area, and the more prominent the shadow feature. When $\beta = 1$, it means that the location is always occluded and does not receive radar echoes, showing a pure shadow area. When $\beta = 0$, it means that the location is never occluded by moving targets, indicating the background clutter area. Corresponding to the two shadow models in Figure 3, the trend of $\beta$ change along the direction of target motion can be classified and discussed. Similarly, $x$ is used to represent the positions of scattering points in the target shadow region along the direction of motion.

(1)  Type I shadow, $L_T \leq L_C$

The shadow coefficient $\beta$ as a piecewise function of $x$ is shown in Equation (9):

$$\beta = \begin{cases} \frac{L_S + 2x}{2L_C}, & -\frac{L_S}{2} \leq x \leq L_T - \frac{L_S}{2} \\ \frac{L_T}{L_C}, & L_T - \frac{L_S}{2} < x \leq L_C - \frac{L_S}{2} \\ \frac{L_S - 2x}{2L_C}, & L_C - \frac{L_S}{2} < x \leq \frac{L_S}{2} \end{cases} \tag{9}$$

Among them, in positions less than the target length, the shadow coefficient increases linearly with the increase of $x$; in the interval greater than the target length but less than the critical size, the shadow coefficient remains unchanged and less than 1; in positions greater than the critical size, the shadow coefficient decreases linearly with the increase of $x$; and the shadow coefficient returns to zero when it exceeds.

(2)  Type II shadows, $L_T > L_C$

The shadow coefficient $\beta$ as a piecewise function of $x$ is shown in Equation (10):

$$\beta = \begin{cases} \frac{L_S + 2x}{2L_C}, & -\frac{L_S}{2} \le x \le L_C - \frac{L_S}{2} \\ 1, & L_C - \frac{L_S}{2} < x \le L_T - \frac{L_S}{2} \\ \frac{L_S - 2x}{2L_C}, & L_T - \frac{L_S}{2} < x \le \frac{L_S}{2} \end{cases} \tag{10}$$

At this time, in the part less than the critical size, the shadow coefficient increases linearly with the increase of $x$; the part above the critical size but less than the target length is the stable shadow area, theoretically not receiving any radar echoes, showing as pure shadow, so its shadow coefficient $\beta$ is equal to 1; in the part above the target length, the shadow coefficient decreases linearly with the increase of $x$; and the shadow coefficient returns to zero when exceeds.

According to $\beta$ and Equations (2) and (3), the ratio of the intensity of the shadow area to the background area can be calculated, denoted as the shadow clutter ratio *ShCR*. The expression of *ShCR* is as follows:

$$ShCR = \frac{\sigma_S}{\sigma_C} = \frac{\sigma_{N} + (1 - \beta)\sigma_b}{\sigma_{N} + \sigma_b} \tag{11}$$

Substituting $\beta$ into Equation (11), the piecewise function of *ShCR* can be obtained as Equations (12) and (13).

(1)  Type I shadows

$$ShCR = \begin{cases} 1 - \frac{(L_S + 2x)\sigma_b}{2L_C(\sigma_N + \sigma_b)}, & -\frac{L_S}{2} \le x \le L_T - \frac{L_S}{2} \\ 1 - \frac{L_T \sigma_b}{L_C(\sigma_N + \sigma_b)}, & L_T - \frac{L_S}{2} < x \le L_C - \frac{L_S}{2} \\ 1 - \frac{(L_S - 2x)\sigma_b}{2L_C(\sigma_N + \sigma_b)}, & L_C - \frac{L_S}{2} < x \le \frac{L_S}{2} \end{cases} \tag{12}$$

(2)  Type II shadows

$$ShCR = \begin{cases} 1 - \frac{(L_S + 2x)\sigma_b}{2L_C(\sigma_N + \sigma_b)}, & -\frac{L_S}{2} \le x \le L_C - \frac{L_S}{2} \\ \frac{\sigma_N}{\sigma_N + \sigma_b}, & L_C - \frac{L_S}{2} < x \le L_T - \frac{L_S}{2} \\ 1 - \frac{(L_S - 2x)\sigma_b}{2L_C(\sigma_N + \sigma_b)}, & L_T - \frac{L_S}{2} < x \le \frac{L_S}{2} \end{cases} \tag{13}$$

It is easy to see that while $ShCR \le 1$, the smaller *ShCR*, the more significant the shadow feature. It is not difficult to analyze the system's detection performance based on *ShCR*.

## 3. Experiment and Analysis of Dynamic Target Shadow Characteristics in ViSAR System

### 3.1. Maximum Detectable Speed of Moving Targets

Furthermore, simulation analysis was conducted on the variation of moving targets in different shadow positions under a typical ViSAR system. The experimental parameters are shown in Table 1. Here, a 0.5-m resolution Ka-band airborne ViSAR system was selected, with a typical target length of 8 m, and *ShCR* of targets moving at different speeds were compared and analyzed.

**Table 1.** Experimental parameters.

| | Parameters | Values |
|---|---|---|
| System parameters | Frequency $f$ (GHz) | 35 |
| | Resolution $\rho$ (m) | 0.5 |
| | Platform altitude $H$ (km) | 600 |
| | Platform velocity $V$ (m/s) | 7556 |
| | Synthetic aperture time $T_a$ (s) | 0.63 |
| | Viewing angle $\alpha$ (°) | 40 |
| Target parameters | Target velocity $V_T$ (m/s) | 3, 6, 9, 12, 15, 18 |
| | Target length $L_T$ (m) | 8 |

Figure 4 shows the results of Experiment 2, displaying the shadow clutter ratio characteristics curves of moving targets with different speeds but the same target size. The dashed line corresponds to $ShCR = -3$ dB. The speeds of the six targets in the figure are 3 m/s, 6 m/s, 9 m/s, 12 m/s, 15 m/s, and 18 m/s. When the target speeds are 9 m/s, 12 m/s, 15 m/s, and 18 m/s, Type I shadows can be formed, with relatively small shadow area sizes at low $ShCR$. Especially when the speed reaches 18 m/s, the center area of the shadow is also higher than $-3$ dB.

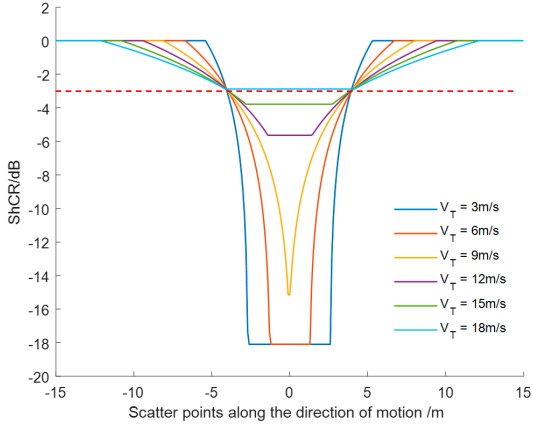

**Figure 4.** Characteristic curve of different-speed moving targets with the same size.

Other targets with lower speeds form Type II shadows, with relatively larger shadow area sizes at low $ShCR$. Under the same conditions, the average intensity of the central region of Type I shadows for the same target is higher than $ShCR$ for Type II shadows of that target. Additionally, it is observed that for a specific system, the characteristic curves of different speed targets intersect near $-3$ dB. When at the center of the shadow, taking according to Equation (12), we can obtain two values, i.e., $x_1 = \frac{\sigma_N + \sigma_b}{2\sigma_b} L_c - \frac{L_S}{2}$ and $x_2 = \frac{L_S}{2} - \frac{\sigma_N + \sigma_b}{2\sigma_b} L_c$, in the gradient region. The length of this interval is $x_2 - x_1 = L_T - \frac{\sigma_N}{\sigma_b} L_c$. When $\frac{\sigma_N}{\sigma_b}$ is high enough, the length of this interval is approximately equal to $L_T$. Therefore, this paper chooses $-3$ dB as the threshold value, which is important for shadow detection of distributed scatterers and can simplify calculations under this condition. For a given system, when the detectable speed threshold is set to $-3$ dB, the maximum detectable speed of moving target shadows can be approximately estimated as Equation (14) according to Equation (12).

$$V_{max} \leq \frac{4\rho V L_T \sigma_b}{\lambda R_0 (\sigma_b + \sigma_N)} \tag{14}$$

The maximum detectable speed can be used to estimate the system's detection capability for moving target shadows. When the speed is below the maximum detectable speed, it can be considered that the target can be detected.

### 3.2. Minimum Clutter Backscatter Coefficient

Next, under the same system, with a typical target speed of 5 m/s, the characteristics of *ShCR* on different sizes were compared and analyzed.

Figure 5 shows the results of Experiment 2, displaying the clutter-to-shadow ratio characteristics curve of targets *ShCR* of different lengths at the same speed. The horizontal axis represents the position coordinates of the shadow area, with the center position of the shadow area as the zero point. The lengths of the six targets in the figure are 2 m, 4 m, 6 m, 8 m, 10 m, and 12 m, respectively.

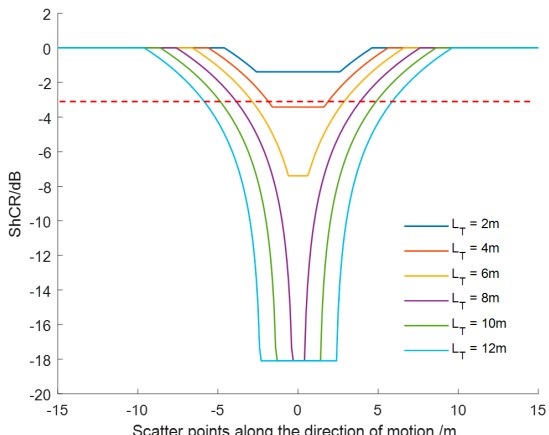

**Figure 5.** Characteristic curves of targets with different sizes at the same velocity.

In Figure 5, the position where the clutter-to-shadow ratio is 0 dB represents the clutter background, and the region where the clutter-to-shadow ratio is below 0 dB is the moving target shadow area, $ShCR = -3$ dB corresponding to the dashed line. Moving targets with lengths of 2 m, 4 m, and 6 m form Type I shadows with relatively smaller shadow area sizes. The three targets with lengths above 6 m form Type II shadows, with relatively larger shadow area sizes. Shorter targets are more likely to form Type I shadows. From Equation (12), it can be seen that the clutter backscatter coefficient significantly affects the calculation of *ShCR*. This is evident when $\sigma_b$ is very small and the ratio of $\sigma_b$ to $\sigma_N$ cannot be ignored (the background clutter is weak), making it difficult to distinguish the shadow of a stationary target from the background. Similar to the maximum detectable speed, under fixed system parameters, it is feasible to assume the existence of a minimum clutter backscatter coefficient that reflects the shadow detection capability of the video SAR system, defined as $\sigma_{b\_min}$. By also taking the critical value of *ShCR* for moving targets, the minimum clutter backscatter coefficient, $\sigma_{b\_min}$, can be determined when there is no pure shadow area in the target shadow.

$$\sigma_{b\_min} = \frac{V_T \lambda R_0 \sigma_N}{4\rho V L_T - V_T \lambda R_0}(\sigma_n < \sigma_b, V_T < V_{max}) \tag{15}$$

Equation (15) implies two conditions: firstly, the target's motion speed must be lower than the maximum detectable velocity, and secondly, the background backscatter coefficient must be higher than the noise-equivalent backscatter coefficient. In the presence of a pure shadow area in the target's shadow, $\sigma_{b\_min} = \sigma_b$. When $\sigma_b$ is greater than the minimum background clutter ratio $\sigma_{b\_min}$, and the target velocity is less than the maximum detectable velocity $V_{max}$, it can be considered that moving target shadows can be effectively detected.

### 3.3. Effective Shadow Pixel Count for Dynamic Targets

In the context of ViSAR area target shadow detection, the number of pixels has a certain impact on detection capability. When the number of pixels in the target shadow is too low, missed detections are highly likely to occur. As the number of pixels increases, the detection probability of the target also increases. However, for large targets with hundreds of connected pixels, increasing the pixel count has a marginal effect on detection performance [18]. Additionally, as mentioned earlier, both the number of pixels in the dynamic target shadow area and the shadow clutter ratio are important parameters affecting ViSAR dynamic target shadow detection.

Considering that the scattering elements in shadow areas with a high shadow clutter ratio make little substantive contribution to shadow detection (such as the regions at both ends of the shadow where the scattering characteristics are similar to clutter), if we set the threshold of the dynamic target shadow detection algorithm as $\varepsilon$ and assume that all pixels with a shadow clutter ratio $ShCR$ are suitable for effective shadow detection, this paper defines the effective shadow pixel count for ViSAR dynamic targets as $N_{SE}$, as shown in Equation (16).

$$N_{SE} = \sum\nolimits_{i \in N_S} k_i, \quad k_i = \begin{cases} 0, & ShCR_i > \varepsilon \\ 1, & ShCR_i \leq \varepsilon \end{cases} \tag{16}$$

where $i$ represents the scattering points in the shadow area. When $k_i = 1$ represents that the $i$ pixel is available for shadow detection, and when $k_i = 0$ represents that the pixel is invalid. For a specific target, if the threshold for dynamic target shadow detection is set at $\varepsilon = -3$ dB, the formula for calculating the effective shadow pixel count can be simplified to Equation (17).

$$N_{SE} = \begin{cases} 0, & ShCR_{ps} > -3 \text{ dB} \\ \left(L_T - \frac{\sigma_N}{\sigma_b} L_c\right) \frac{w_s}{\rho \rho_r}, & ShCR_{ps} \leq -3 \text{ dB} \end{cases} \tag{17}$$

In Equation (17), $\rho_r$ represents the range resolution. Where $ShCR_{ps}$ represents $ShCR$ on the center of the shadow. The effective shadow pixel count for moving targets $N_{SE}$ can comprehensively reflect the impact of the number of pixels in the moving target shadow area and the parameter $ShCR$ on the system's shadow detection performance.

### 3.4. Analysis of Typical ViSAR System Moving Target Shadow Characteristics

To validate the effectiveness of $N_{SE}$, this section compares and analyzes the dynamic target shadow characteristics of three airborne ViSAR systems and two spaceborne ViSAR systems using the Hamas rocket launcher as a typical moving target. The length, width, and height of the Hamas rocket launcher are 7 m, 2.4 m, and 3.2 m, respectively, with a maximum traveling speed of 85 km/h. The real-time detection and tracking capabilities of ViSAR are influenced by the imaging resolution and frame rate [7,19]. Additionally, imaging with aperture overlap inevitably requires longer accumulation times, leading to some imaging delay and poor real-time performance, which may not meet the requirements for real-time monitoring of imaging scenes [20]. As shown in Table 2, we selected parameters for five typical spaceborne and airborne ViSAR systems, with a target velocity set at 10 m/s, using dry soil in the Ka band as the background. The scene parameters were set as $\sigma_b = \overline{\sigma_b} - 14.8$ dB, $\sigma_n = -48.7$ dB, and $MNR = -18.2$ dB. Meanwhile, we analyzed and calculated various dynamic target shadow characteristic parameters, such as critical size, shadow pixel count, and effective pixel count.

From Table 3, it can be observed that ViSAR systems a1 and s2 can form Type II shadows containing pure shadow areas, while systems a2, a3, and s1 can only form Type I shadows. In system a1, the effective pixel count and the length of the pure shadow area are much higher than in other systems, indicating that it is the system with the best performance in moving target shadow detection. In system a2, the number of moving target shadow pixels is large, but the effective pixel count is 0. This is because its larger synthetic aperture time results in a larger shadow area, but at the same time, its larger critical target size leads to a high clutter-to-shadow ratio, making effective shadow detection

difficult. Among the other three systems, the number of shadow pixels is equal. Taking into account the contribution of the clutter-to-shadow ratio, these systems should have similar performance in moving target shadow detection.

**Table 2.** Experimental parameters.

| | Parameters | Values |
|---|---|---|
| System parameters | Center frequency $f$ (GHz) | 35 |
| | Resolution $\rho$ (m) | 0.5 |
| | Platform altitude $H$ (km) | 600 |
| | Platform velocity $V$ (m/s) | 7556 |
| | Synthetic aperture time $T_a$ (s) | 0.63 |
| | Viewing angle $\alpha$ (°) | 40 |
| Target parameters | Target velocity $V_T$ (m/s) | 10 |
| | Target length $L_T$ (m) | 2, 4, 6, 8, 10, 12 |

**Table 3.** Shadow characteristic parameters of Hamas rocket launchers in airborne/spaceborne ViSAR systems.

| | Parameters | Airborne System a1 | Airborne System a2 | Airborne System a3 | Spaceborne System s1 | Spaceborne System s2 |
|---|---|---|---|---|---|---|
| System parameters | Frequency $f$ (GHz) | 235 | 35 | 35 | 35 | 35 |
| | Resolution $\rho$ (m) | 0.2 | 0.5 | 0.5 | 0.5 | 0.5 |
| | Platform altitude $H$ (km) | 4 | 8 | 8 | 600 | 400 |
| | Platform velocity $V$ (m/s) | 80 | 40 | 80 | 7556 | 7667 |
| | Synthetic aperture time $T_a$ (s) | 0.23 | 2.42 | 1.21 | 0.96 | 0.63 |
| | Incident angle $\alpha$ (°) | | | 40 | | |
| Target parameters | Target velocity $V_T$ (m/s) | | | 10 | | |
| | Critical size $L_C$ (m) | 2.1 | 22.3 | 11.2 | 8.9 | 5.8 |
| | Shadow area length $L_S$ (m) | 9.1 | 29.3 | 18.2 | 15.9 | 12.8 |
| | Shadow area width $W_S$ (m) | 5.6 | 5.6 | 5.6 | 5.6 | 5.6 |
| | Pure shadow length $L_{PS}$ (m) | 4.5 | 0 | 0 | 0 | 0.2 |
| | Shadow area pixel count $N_{SE}$ | 1277 | 657 | 407 | 355 | 287 |
| | Effective shadow pixel count $\overline{N_{SE}}$ | 392 | 0 | 154 | 154 | 154 |

Furthermore, for the above five sets of ViSAR system parameters, different speed-moving target shadow images were simulated, as shown in Figure 6. Figure 6a shows a schematic of the scene target settings, with targets 1 to 6 moving at speeds of 3 m/s, 6 m/s, 9 m/s, 12 m/s, 15 m/s, and 18 m/s from left to right, respectively, with blank areas representing uniform background clutter. Figure 6b–f display the SAR images of moving target shadows at different speeds for the five ViSAR systems, visually reflecting the differences and trends in shadow sizes and clutter-to-shadow ratios of each target. The shadow size of the moving targets in the SAR images of each ViSAR system increases with the target speed, while the shadow grayscale values gradually approach the background clutter grayscale values, making them less distinguishable.

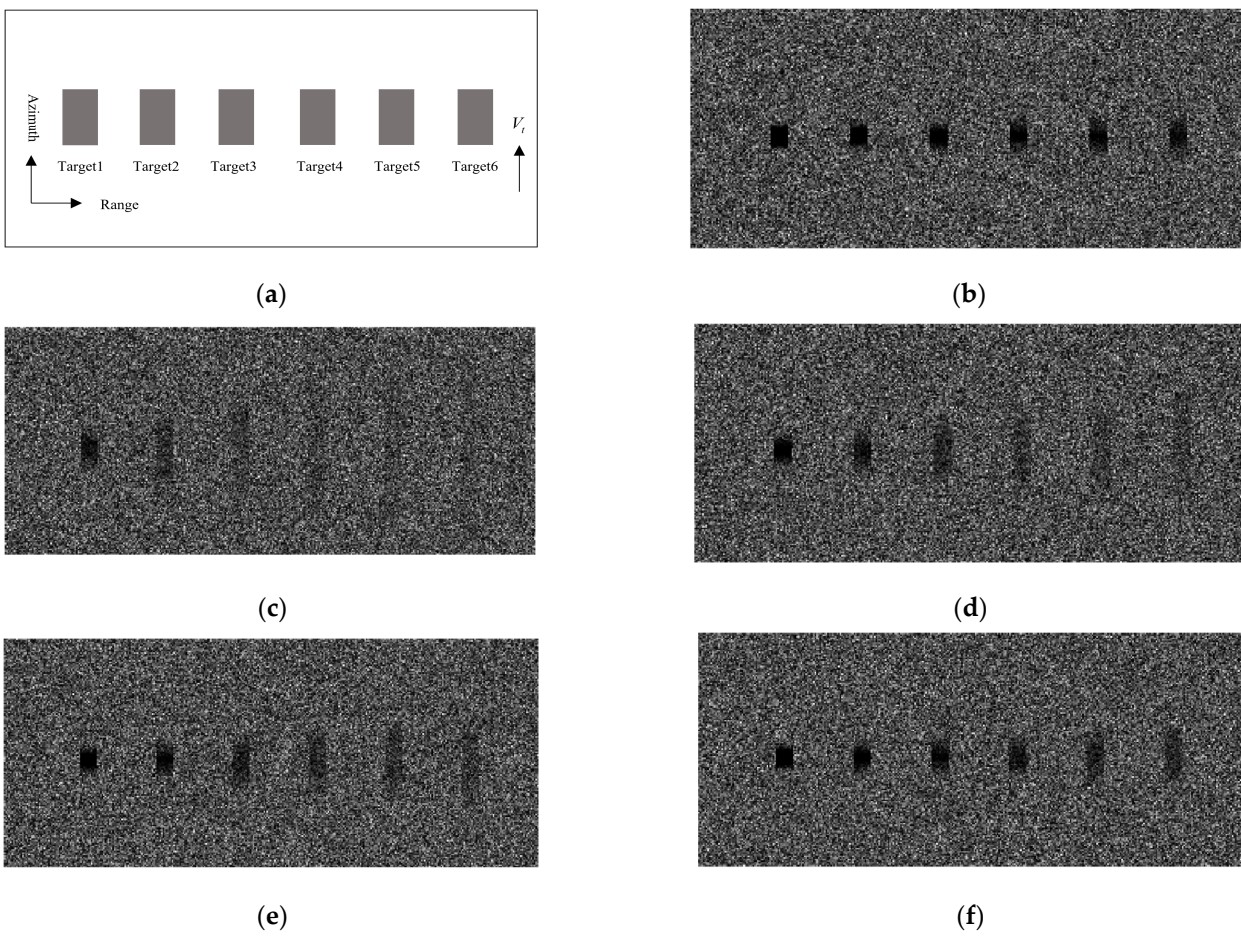

**Figure 6.** Target shadows under typical system parameters. (**a**) Schematic of scene target settings. (**b**) Shadow image of airborne system a1. (**c**) Shadow image of airborne system a2. (**d**) Shadow image of airborne system a3. (**e**) Shadow image of spaceborne system s1. (**f**) Shadow image of spaceborne system s2.

Figure 7a–f present the shadow clutter ratio curves of dynamic target shadows for five typical ViSAR systems under different target velocities, respectively. The horizontal axis in each figure represents scattering points along the direction of motion, measured in meters.

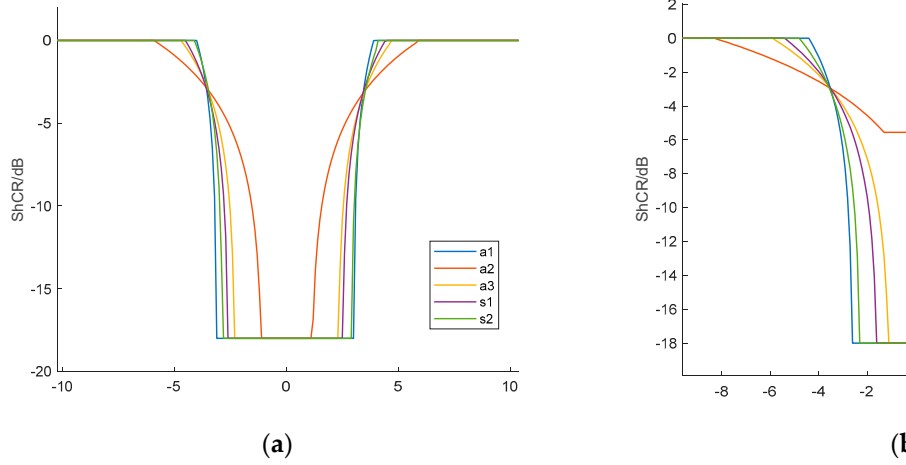

**Figure 7.** *Cont.*

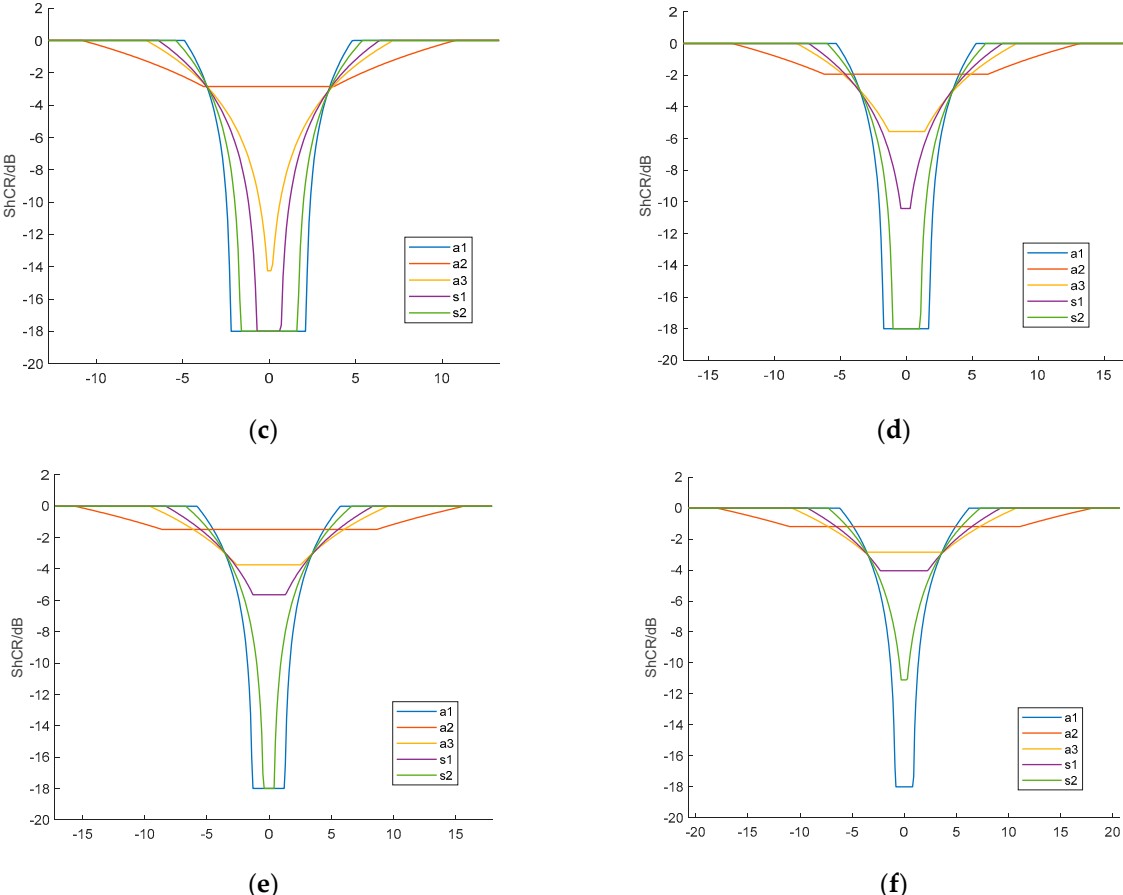

**Figure 7.** Trend of clutter-to-shadow ratio (*ShCR*) for targets at different speeds. (**a**) $V_T$ = 3 m/s. (**b**) $V_T$ = 6 m/s. (**c**) $V_T$ = 9 m/s. (**d**) $V_T$ = 12 m/s. (**e**) $V_T$ = 15 m/s. (**f**) $V_T$ = 18 m/s.

In the meantime, Table 4 provides the effective pixel counts of dynamic target shadows under different scenarios, where $\overline{N_{SE}}$ represents the average effective shadow pixel count at different velocities.

**Table 4.** Effective pixel count of moving target shadows for different systems.

| ViSAR System | 3 | 6 | 9 | 12 | 15 | 18 | $\overline{N_{SE}}$ |
|:---:|:---:|:---:|:---:|:---:|:---:|:---:|:---:|
| a1 | 392 | 392 | 392 | 392 | 392 | 392 | 392 |
| a2 | 154 | 154 | 0 | 0 | 0 | 0 | 51 |
| a3 | 154 | 154 | 154 | 154 | 0 | 0 | 103 |
| s1 | 154 | 154 | 154 | 154 | 154 | 0 | 128 |
| s2 | 154 | 154 | 154 | 154 | 154 | 154 | 154 |

Meanwhile, we calculated the effective pixel counts of shadows for targets of different sizes, using typical dimensions for small vehicles (length 3.8 m/width 1.5 m/height 1.5 m), medium-sized vehicles (5 m × 1.8 m × 1.6 m), and large vehicles (7 m × 2.4 m × 3.2 m) as examples under each system. As shown in Table 5.

**Table 5.** Effective pixel count of moving target shadows for targets of different sizes.

| ViSAR System | Small Vehicles | Medium-Sized Vehicles | Large Vehicles | $\overline{N_{SE}}$ |
|:---:|:---:|:---:|:---:|:---:|
| a1 | 114 | 170 | 392 | 232 |
| a2 | 0 | 0 | 0 | 0 |
| a3 | 0 | 72 | 154 | 78 |
| s1 | 0 | 72 | 154 | 78 |
| s2 | 44 | 72 | 154 | 93 |

The results of the comprehensive analysis of Figures 6 and 7, as well as Tables 3–5, show that the grayscale differences of moving target shadows in SAR images are consistent with the shadow clutter ratio curve. It can be considered that the ability of various ViSAR systems to detect moving target shadows from high to low is a1 > s2 > s1 > a3 > a2. (1) The airborne THz ViSAR system a1 in Figure 6b has the most significant features of moving targets, and the changes in target motion speed are relatively small. This system should have the best performance in detecting moving target shadows.

(2) The two airborne Ka-band ViSAR systems a2 and a3 have platform speeds of $V(a3) > V(a2)$. System a2 can form more significant shadow features for targets with speeds less than 6 m/s, while system a3 can form more significant shadow features for targets with speeds less than 12 m/s. Moving targets with higher speeds will be difficult to detect effectively. Overall, the airborne system a3 with higher platform speed is better than system a2.

(3) For the two spaceborne Ka-band ViSAR systems s1 and s2, with orbit heights $H(s1) > H(s2)$, when the target speed does not exceed 15 m/s, the number of effective shadow pixels formed by system s1 is slightly higher than that of system s2. This is because the higher orbit height of system s1 results in a larger synthetic aperture time, critical dimension, and shadow area. When the target speed is 18 m/s, although system s1 theoretically can form a larger shadow area, the number of effective shadow pixels decreases significantly due to the high shadow clutter ratio, as shown by the shadows in Figure 6f being very close to the clutter background and difficult to distinguish. Overall, the spaceborne system s2 with a lower orbit height is more suitable for detecting the shadows of higher-speed moving targets.

In conclusion, using a higher radar frequency band, increasing the speed of airborne platforms, or appropriately reducing the orbit height of spaceborne systems all have the potential to obtain more significant shadow features of high-speed moving targets.

## 4. ViSAR Moving Target Shadow Detection Simulation Experiment

### 4.1. Experimental Method

Researchers both domestically and internationally have proposed three main types of methods for ViSAR dynamic target detection and tracking. The first method involves initially detecting shadow areas in each frame of video images using grayscale features, followed by data association and target tracking on the temporal detection results. This method accomplishes false alarm suppression and motion state estimation, outputting two-dimensional position and velocity parameters of the target. However, it has relatively low computational complexity and performance [21]. The second method involves detecting moving target shadows using information from frame differencing between consecutive frames, followed by dynamic target tracking. While this method offers better performance, it is more computationally complex [5]. The third method involves extracting background information by averaging multiple frames and then using background subtraction to detect and track moving targets. Although this method also has higher complexity and provides some performance improvement, it does not work well for backgrounds with multiple target movements [16]. Considering the crucial role of grayscale features in dynamic target shadow detection and the role of sequence information in suppressing false alarms of

stationary target shadows during the target tracking stage, this paper adopts the first method. It focuses on verifying the detection of dynamic target shadows in single-frame SAR images under uniform clutter backgrounds. The experimental design is illustrated in Figure 8.

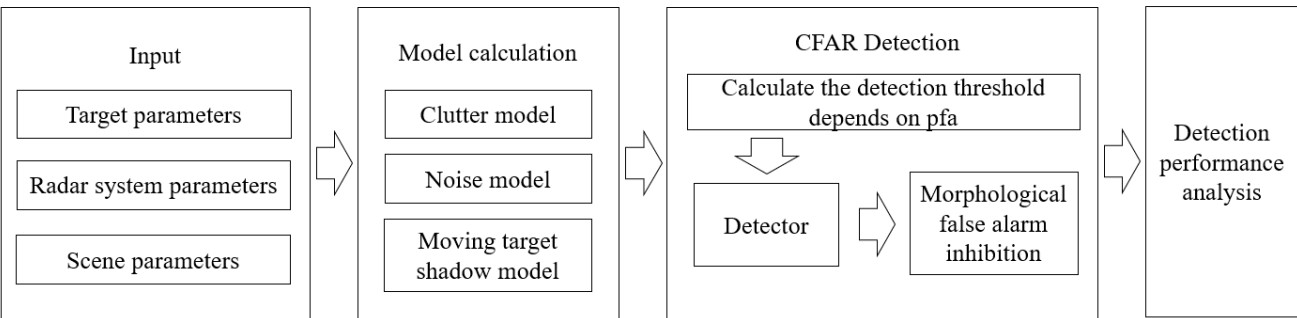

**Figure 8.** Experiment design approach.

First, target parameters, radar system parameters, and scene parameters are used as inputs to simulate clutter and shadows in ViSAR images separately. Then, the normalized detection threshold for CFAR experiments is calculated based on the false alarm probability of each pixel. Next, the ViSAR image sequence is input into the detector for shadow detection, and false alarms are suppressed through morphological filtering. Finally, the detection results are statistically analyzed to evaluate the detection performance.

According to [22,23], the phase of scatterers in SAR images is randomly distributed between $0 \sim 2\pi$, and the real and imaginary parts of the composite echo are statistically independent zero-mean Gaussian random variables. The amplitude follows a Rayleigh distribution, and the intensity follows a negative exponential distribution. The probability density function of intensity is as shown in Equation (18).

$$f(I) = \frac{1}{\langle I \rangle} e^{-\frac{1}{\langle I \rangle}} \tag{18}$$

$I$ is the intensity of a pixel point in an image. For a uniform clutter background, the average intensity of the pixel points is equal to the standard deviation of the speckle intensity [24], that is $\langle I \rangle = \sqrt{var(I)}$.

Unlike traditional detection tasks, shadow detection mainly focuses on detecting regions below the detection threshold. The noise distribution in the integral below the threshold corresponds to the detection probability at that point; the integral above the threshold corresponds to the missed detection probability; and the integral of the clutter distribution below the threshold corresponds to the false alarm probability. Therefore, the expression for the false alarm probability of a pixel point $P_{fa}$ can be obtained as shown in Equation (19).

$$P_{fa} = \int_0^\varepsilon f_{IB}(x)dx = 1 - e^{-\frac{\varepsilon}{\sigma_C}} \tag{19}$$

In Equation (19), $f_{IB}$ is the probability density function of the clutter area, and $\varepsilon$ is the detection threshold. The detection threshold can be calculated based on the false alarm probability.

The above analysis is based on point scatterers. Next, simulation detection experiments will be conducted to analyze the detection performance of moving target shadow face targets. In the experiment, the number of correct detection results is defined as $tp$, the number of false detection results is defined as $fp$, and the number of missed detection targets is defined as $fn$. During the experiment, we statistically analyzed the connected regions detected. If the centroid of a connected region falls within the actual shadow area of the scene, we consider it a correct detection result. Similarly, if the centroid falls outside the actual shadow area of the target, we consider it a false detection result. Finally, we tally

the number of targets that have shadows but no centroids, which are defined as missed detection results. Subsequently, the detection rate of shadows $P_D$ and the false alarm rate $P_{FA}$ are defined as shown in Equation (20).

$$P_D = \frac{tp}{tp + fn} \tag{20}$$

$$P_{FA} = \frac{fp}{tp + fn + fp}$$

### 4.2. Analysis of Experimental Results

For the detection of connected shadow regions, ViSAR dynamic target shadows employ morphological filtering and density clustering as false alarm suppression methods, but they also require reliable *ShCR* to achieve effective detection. To suppress false alarm rates, we employ disk-shaped matrix morphological closing operations with a radius of 2 pixels and hole filling, followed by erosion operations with a disk-shaped matrix of radius 1 to process the detection results based on the actual size of the Hamas target. Finally, connected component analysis is performed to obtain the desired targets and statistics [25]. By selecting different single-point false alarm probabilities, we obtain the curve of Hamas area target detection rate versus shadow clutter ratio. We chose $\sigma_b = \overline{\sigma_b} - 16.5$ dB. Based on this, 6000 Monte Carlo simulation experiments were conducted. The experimental results are displayed in Figure 9.

Figure 9a–c correspond to detection rate versus false alarm rate curves for $SHCR = -3$ dB of $P_{fa} = 0.15, 0.2$, and $0.25$. The dashed lines in the figures intersect the horizontal axis at the speeds corresponding to the center positions of the shadow areas at this threshold. It is evident that targets with speeds below this threshold can achieve robust detection, and the detection performance declines as the target speed exceeds this threshold. As $P_{fa}$ increases, the range of detectable speeds for targets gradually expands, the false alarm rate for area targets also increases, stabilizing initially around $10^{-4}$ and subsequently stabilizing around $1.5 \times 10^{-2}$. Taking system s1 as an example, the detection rate starts to decrease after the speed exceeds 8 m/s without generating false alarms at $P_{fa} = 0.15$. At $P_{fa} = 0.2$, this speed is approximately 9 m/s, and at $P_{fa} = 0.25$, it is around 10 m/s, all near the speeds corresponding to the $-3$ dB point. Therefore, it can be considered that $SHCR$ can provide an approximate estimate of the system's detection performance for moving targets.

Through the analysis of Equation (14), it is evident that $\sigma_b$ also affects the detection performance. Therefore, under the parameters of system s2, simulations and detections of dynamic target shadows were conducted for soil ($\sigma_b = -14.8$ dB), grass ($\sigma_b = -21.4$ dB), and road surfaces ($\sigma_b = -28.5$ dB) in the Ka band, as shown in Figure 10. The dashed lines intersecting the horizontal axis represent the speeds corresponding to the center positions of the shadow areas at $ShCR = -3$ dB. From Figure 9a, it can be observed that as $\sigma_b$ increases, the range of detectable speeds for targets also increases. However, when the speed becomes too high, the number of effective pixel counts becomes insufficient to support detection. Figure 9b shows the corresponding false alarm rate curves. A higher $\sigma_b$ corresponds to a larger maximum detectable speed, which is consistent with the conclusion of Equation (14).

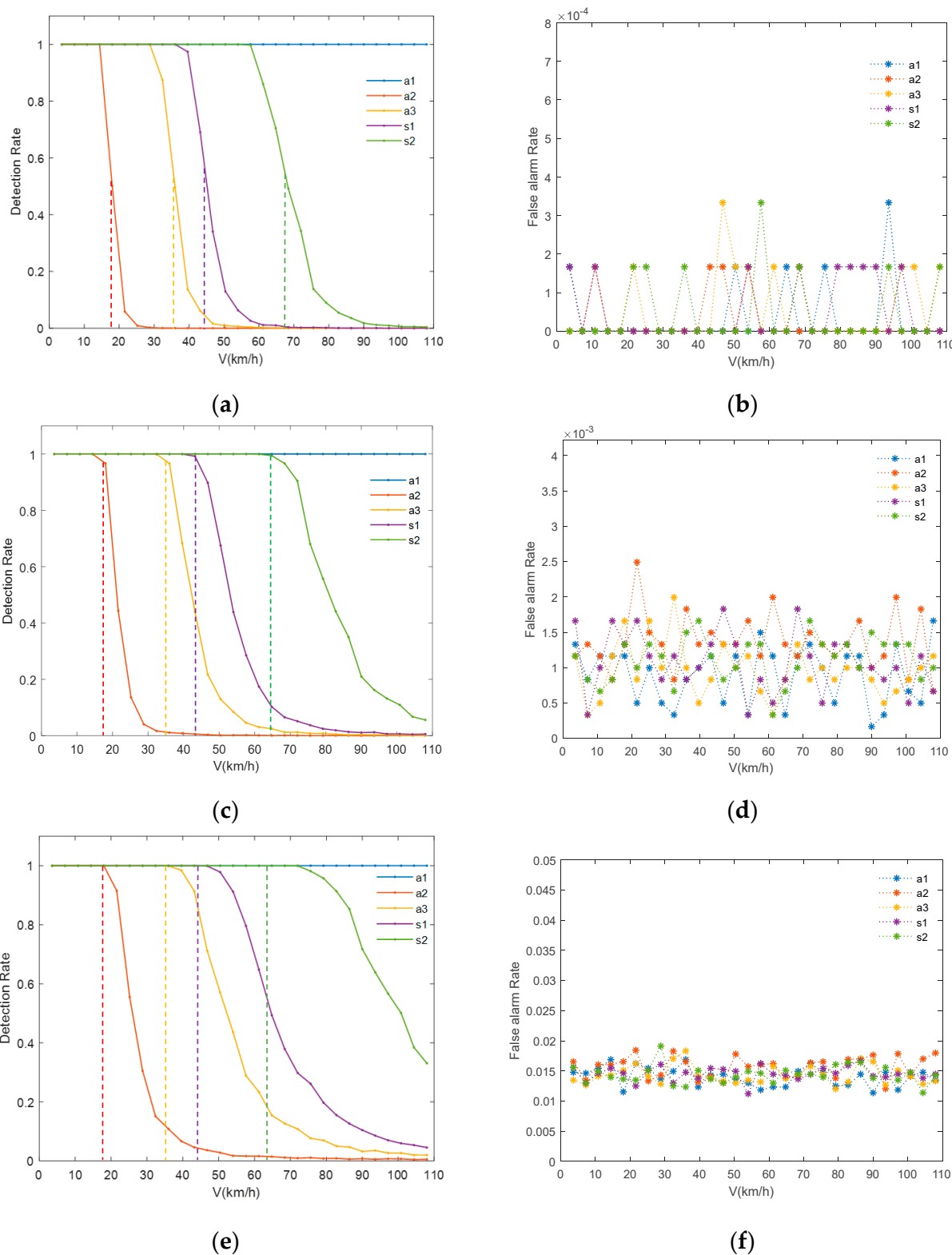

**Figure 9.** Experimental results of target detection under different false alarm rates. (**a**,**b**) The detection rate curve and false alarm rate curve ($P_{fa} = 0.15$). (**c**,**d**) The detection rate curve and false alarm rate curve ($P_{fa} = 0.2$). (**e**,**f**) The detection rate curve and false alarm rate curve ($P_{fa} = 0.25$).

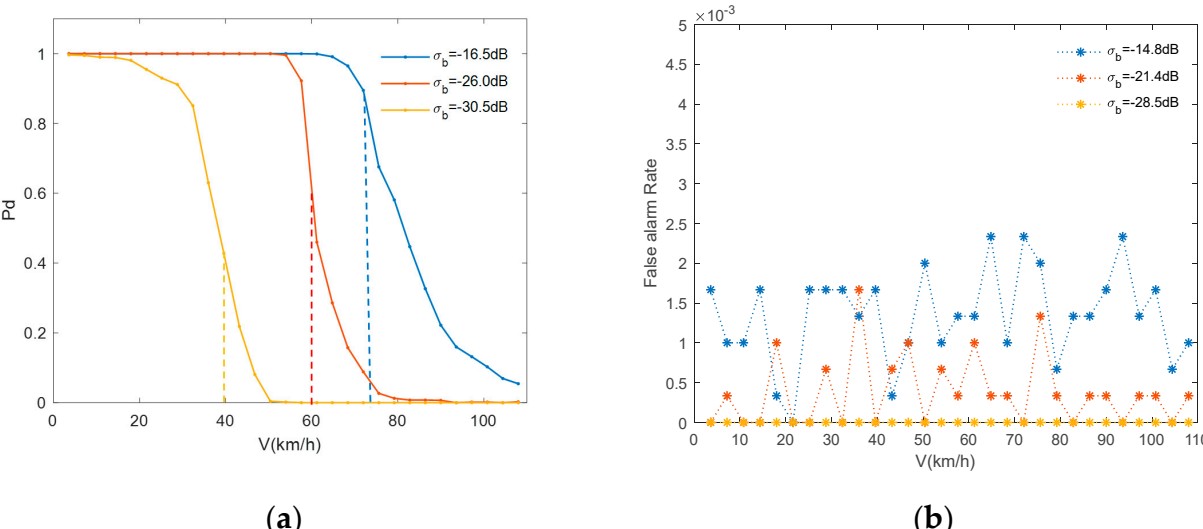

**Figure 10.** Detection performance of target shadows in different scenes ($P_{fa} = 0.2$). (**a**) Detection rate curve. (**b**) False alarm rate curve.

## 5. Discussion

There is a discussion of some details in this article. Firstly, we found that the radar incidence angle undoubtedly affects the target shadows. According to the theory in this paper, changing the incident angle will, on the one hand, alter the surface's equivalent backscatter coefficient and, on the other hand, change the area of the shadow region. In Section 4.2, this paper analyzes the impact of the equivalent backscatter coefficient on the detection performance of the ViSAR system. Increasing the incident angle can enlarge the shadow area, which is beneficial for improving the detection probability. However, it typically leads to a decrease in the equivalent backscatter coefficient of the scene, making it more difficult to distinguish between target shadows and the background. Additionally, for high-resolution ViSAR systems, there is a boundary effect of the shadow region size on the detection performance of target shadows. When the shadow region size is too large, the shadow detection probability will not significantly change with this parameter.

Then, we found that the direction of motion does have a certain impact on whether the shadow of the target can be detected, but due to the phenomena of energy diffusion and offset in moving targets, even small velocity components in the range direction can cause the target's energy to shift away from its actual position. Therefore, all shadows in this paper are simulated under the condition that the energy shift of the target does not overlap with the shadow. When the target strictly moves along the azimuth, the shadow will no longer be significant enough. However, due to the elevation obstruction of the target itself, a shadow area with a certain area can still be formed. The shadow characteristics of this shadow area can be used in the detection task of target shadows.

In this paper, the shadows of targets moving in different directions are approximated as rectangular shadow regions, as shown in Figure 2b, without considering the influence of irregular objects. After analysis, we believe that for the method proposed in this paper, irregular shapes do not significantly affect the detection performance. This is because the main statistics in this paper are the number of shadow pixels that meet the *ShCR* condition without imposing restrictions on the morphology of the detection results. If irregular motion target shadows need to be filtered out to suppress false alarms based on the morphology of the detection results, it may affect the detection capability of the ViSAR system.

Finally, during the experimental process, we realized that distinguishing shadows from low-scattering areas in ViSAR images remains challenging, potentially leading to false alarms when using shadow intensity detection methods. To better overcome this challenge,

contrast enhancement between shadows and low-scattering areas can be attempted. Additionally, improving algorithms to obtain more shadow information, such as temporal information and shadow edge information, can enhance shadow detection capabilities. For scenes with weak background scattering, such as water bodies, relying solely on shadow information to perceive target status is challenging, which may require us to analyze the feasibility of target detection from other perspectives.

## 6. Conclusions

In conclusion, based on the imaging mechanism of dynamic target shadows in ViSAR, this paper proposes concepts such as critical size, shadow clutter ratio, and effective pixel count. It quantitatively analyzes the formation mechanism and detection performance of dynamic target shadows in ViSAR, concluding that the detectability of dynamic target shadows is related to target parameters, system parameters, and background parameters. The paper also provides the maximum detectable speed of shadow targets, the minimum clutter-to-backscatter ratio, and a method for evaluating the detection performance of dynamic target shadows by calculating their effective pixel count under certain conditions. Experimental detection of ViSAR dynamic target shadows under different system and scene parameters validates the conclusions. The paper's approach and conclusions could support the design of ViSAR system parameters.

**Author Contributions:** All the authors made significant contributions to the work. B.W., A.Y. and W.T. designed the research and analyzed the results. methodology, B.W., A.Y. and Z.H. performed the experiments. B.W. and A.Y. wrote the paper. Z.H. and W.T. provided suggestions for the preparation and revision of the paper. All authors have read and agreed to the published version of the manuscript.

**Funding:** This research received no external funding.

**Data Availability Statement:** The data presented in this study are available on request from the corresponding author.

**Conflicts of Interest:** The authors declare no conflict of interest.

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
