# Peer review of "Performance Analysis of Moving Target Shadow Detection in Video SAR Systems"

_remotesensing, doi:10.3390/rs16111825_

Round 1

Reviewer 1 Report

Comments and Suggestions for Authors

This article establishes a ground moving target shadow model for ViSAR, analyzes the effects of SAR system parameters, target, and scene parameters on shadow detection, conducts model level and signal level simulation experiments, and analyzes and obtains the maximum detectable speed of moving targets and minimum clutter backscatter coefficient for ViSAR moving targets. The conclusions of this paper can play an important supporting role in the optimal design of airborne and spaceborne ViSAR systems and the research of high-performance moving target detection algorithms.

The following parts still need further improvement:

 1) Further supplement the commonly used performance evaluation methods for ViSAR moving target shadow detection in the introduction.

 2) In Figure 2, the green and yellow lines can be bolded appropriately. Explain that when there is distance motion between Target 2 and Target 3, the imaging results of the moving target will be defocused and the position will shift, leaving behind shadow features caused by target occlusion.

 3) Supplement the meanings of Lc and LT in Formula 2.

 4) In formula (4),  should be the azimuth resolution.

 5) “surface target” should be “area target” or “extended target”.

 6) Clarify the meaning of MNR in formula (6).

 7) There are multiple spaces and non-standard formulas, such as 197, 222, 225, 257, 271, etc.

 8) In Figure 4, the ShCR label should be consistent with formula (13), which is SHCR. The lowest value of the curve is -18dB, which is mainly determined by what factors?

 9) How is detectable defined for area targets in formula (20)?

Author Response

Hello esteemed Reviewer,

Thank you for your valuable feedback and inquiries. I have enclosed the review report for your perusal in the attachment. Kindly review it at your earliest convenience.

Best regards,

Reviewer 2 Report

Comments and Suggestions for Authors

The paper analyzes the identification of target objects through shadow identification in Video Synthetic Aperture Radar. The proposed theoretical models are validated with simulations. However, in the paper, the proposed models are not validated with measurements and no measured data are presented.

Please find below some comments

- figure 1 could be improved in resolution to improve readability

- in line 95 you mention Figure 2 (c) that is not present

- MNR is not defined

- line 199: “HH polarization mode is higher than that in the same conditions for VV and HV polarization modes”. Is this a property of your simulated data? Please clarify this point

- the caption and the normal text are too close sometimes (e.g., see line 259 in Figure 4 caption)

-in line 4.2 the section is called “ Figures, Tables and Schemes”. I suggest to use a more clear title for the section

-line 474: “ radius of 2” but the unit of measurement is missing

Thanks

Best regards

Comments on the Quality of English Language

none

Author Response

(The authors gave the same response as above.)

Reviewer 3 Report

Comments and Suggestions for Authors

Review of “Performance Analysis of Moving Target Shadow Detection in Video SAR Systems” by Wei et al.

The paper discuss the intricacies of detecting targets or objects from Video Synthetic Aperture Radar (ViSAR). By making simulations with ideal ground-based target and typical sensor parameters the authors were able to come up with limitations on target size, target velocity, sensor velocity and background noise parameters for future sensor/experiment design.

I appreciate the authors effort in bringing out the study. I have a few questions that need to be addressed for clarity.

1. Though it is understood that the influence of the radar look angle is inherent in the critical target length, Can one adjust the look angle of the radar to detect desirable targets? Please evaluate this for a constant target size and velocity.

2. Please illustrate the influence of direction of motion. Is it possible to detect the targets, if the target motion is purely along range or purely along azimuth.

3. What is the minimum size of overlap between the shadows to detect targets that are inseparable. Does this have limitations in range direction separation and azimuth direction separation.

4. What is the influence of irregularly shaped objects or targets and the relation between target shape and look/incidence angle of the radar.

Minor correction:

In line 267-268: “Type I shadows have higher at the center compared to Type II shadows.” Please explain what is ‘higher’ here.

Author Response

(The authors gave the same response as above.)

Reviewer 4 Report

Comments and Suggestions for Authors

In this paper, the authors proposes concepts such as critical size, shadow clutter ratio, and effective pixel count in the ViSAR, and quantitatively analyzes the formation mechanism and detection performance of dynamic target shadows in ViSAR, concluding that the detectability of dynamic target shadows is related to target parameters, system parameters, and background parameters.

The analysis and derivation process of this paper seems reasonable, and the experimental results show that the method has certain advantages, which provides a valuable reference for the research of VISAR target monitoring.

Comments on the Quality of English Language

Minor editing of English language required

Author Response

(The authors gave the same response as above.)

Reviewer 5 Report

Comments and Suggestions for Authors

This paper addresses the modeling of moving target shadows in ViSAR (Video Synthetic Aperture Radar) systems. By analyzing the detection performance of these shadows, the paper aims to contribute to system optimization and evaluation. The research encompasses establishing shadow models, deriving key parameters for shadow detection, analyzing ViSAR system characteristics, and validating the proposed methods through simulated experiments.

The paper recognizes the ongoing challenge of differentiating shadows from low-scattering areas in ViSAR images. This distinction difficulty can potentially lead to false alarms when using shadow intensity detection methods. The authors should describe potential lines of research to address this issue.

The results are promising for future implementation in real, non-simulated systems.

Author Response

(The authors gave the same response as above.)

Round 2

Reviewer 3 Report

Comments and Suggestions for Authors

I am satisfied with the answers provided by the authors, thank you.

However, I suggest the authors to include a discussion section in the paper, with some of the replies provided (I suggest replies to my comment 2 and 4). Inclusion of a discussion section can make a strong paper and provide insights on potential limitations of experiments.

Author Response

(The authors gave the same response as above.)
